# Development and validation of COEWS (COVID-19 Early Warning Score) for hospitalized COVID-19 with laboratory features: A multicontinental retrospective study

Riku Klén[1†], Ivan A Huespe[2*†], Felipe Aníbal Gregalio[2],
Antonio Lalueza Lalueza Blanco[3], Miguel Pedrera Jimenez[3], Noelia Garcia Barrio[3],
Pascual Ruben Valdez[4], Matias A Mirofsky[5], Bruno Boietti[2],
Ricardo Gómez-Huelgas[6], José Manuel Casas-Rojo[7], Juan Miguel Antón-Santos[7],
Javier Alberto Pollan[2], David Gómez-Varela[8*]

[1]Turku PET Centre, University of Turku and Turku University Hospital, Turku,
Finland; [2]Italian Hospital of Buenos Aires, Buenos Aires, Argentina; [3]12 de Octubre
University Hospital, Research Institute of Hospital 12 de Octubre (imas+12),
Complutense University, Madrid, Spain; [4]Vélez Sarsfield Hospital, Buenos Aires,
Argentina; [5]Hospital Municipal de Agudos Dr Leónidas Lucero, Bahía Blanca,
Argentina; [6]Regional University Hospital of Málaga, Biomedical Research Institute
of Málaga (IBIMA), University of Malaga, Málaga, Spain; [7]Infanta Cristina University
Hospital, Madrid, Spain; [8]Division of Pharmacology & Toxicology, Department of
Pharmaceutical Sciences, University of Vienna, Vienna, Austria

*For correspondence:
ivan.huespe@hospitalitaliano.
org.ar (IAH);
david.gomez.varela@univie.ac.
at (DG-V)

†These authors contributed
equally to this work

Competing interest: The authors
declare that no competing
interests exist.

Reviewing Editor: Evangelos J
Giamarellos-Bourboulis, National
and Kapodistrian University of
Athens, Medical School, Greece

## Abstract

**Background:** The emergence of new SARS-CoV-2 variants with significant immune-evasiveness,
the relaxation of measures for reducing the number of infections, the waning of immune protection
(particularly in high-risk population groups), and the low uptake of new vaccine boosters, forecast
new waves of hospitalizations and admission to intensive care units. There is an urgent need for
easily implementable and clinically effective Early Warning Scores (EWSs) that can predict the risk
of complications within the next 24–48 hr. Although EWSs have been used in the evaluation of
COVID-19 patients, there are several clinical limitations to their use. Moreover, no models have been
tested on geographically distinct populations or population groups with varying levels of immune
protection.

**Methods:** We developed and validated COVID-19 Early Warning Score (COEWS), an EWS that is
automatically calculated solely from laboratory parameters that are widely available and affordable.
We benchmarked COEWS against the widely used NEWS2. We also evaluated the predictive perfor-
mance of vaccinated and unvaccinated patients.

**Results:** The variables of the COEWS predictive model were selected based on their predictive coef-
ficients and on the wide availability of these laboratory variables. The final model included complete
blood count, blood glucose, and oxygen saturation features. To make COEWS more actionable in
real clinical situations, we transformed the predictive coefficients of the COEWS model into indi-
vidual scores for each selected feature. The global score serves as an easy-to-calculate measure
indicating the risk of a patient developing the combined outcome of mechanical ventilation or death
within the next 48 hr.

The discrimination in the external validation cohort was 0.743 (95% confidence interval [CI]: 0.703–0.784) for the COEWS score performed with coefficients and 0.700 (95% CI: 0.654–0.745) for the COEWS performed with scores. The area under the receiver operating characteristic curve (AUROC) was similar in vaccinated and unvaccinated patients. Additionally, we observed that the AUROC of the NEWS2 was 0.677 (95% CI: 0.601–0.752) in vaccinated patients and 0.648 (95% CI: 0.608–0.689) in unvaccinated patients.

**Conclusions:** The COEWS score predicts death or MV within the next 48 hr based on routine and widely available laboratory measurements. The extensive external validation, its high performance, its ease of use, and its positive benchmark in comparison with the widely used NEWS2 position COEWS as a new reference tool for assisting clinical decisions and improving patient care in the upcoming pandemic waves.

**Funding:** University of Vienna.

## Editor's evaluation

This is an important contribution where the authors describe the development of a new Early Warning System for patients with COVID-19, namely COEWS. In their convincing approach, they integrate sex, SpO2, complete blood count and blood glucose. A score equal to or higher than 6 represents a high risk for the admitted COVID-19 patient in need of oxygen to be intubated or die within 48 hours. The score is tested in a Spanish cohort and validated through a smaller Spanish cohort and a larger Argentinian one. It shows good performance, better than that of NEWS2. Hence the paper is relevant for clinicians.

## Introduction

The appearance of new SARS-CoV-2 variants with significant immune-evasiveness, the relaxation of measurements for reducing the number of infections, the waning of immune protection (particularly in high-risk population groups), and the low uptake of new vaccine boosters, forecast new waves of hospitalizations and admission in intensive care units (ICUs). This situation will add up to a load of work accumulated and the high pressure supported by emergency departments.

In the midst of this rapidly evolving situation, there is an urgent need for easily implementable and clinically effective decision tools to assist healthcare personnel in their decision-making process. These tools play a crucial role in optimizing the level of care and ensuring the appropriate allocation of resources. In response to this need, numerous predictive models for COVID-19 have been published (*Shakeel et al., 2021*), utilizing vital signs and laboratory such as the PRIORITY (*Martínez Lacalzada et al., 2021*), or only laboratory results such as the COvid-19 Disease Outcome Predictor (CODOP) (*Klén et al., 2022*). However, it is important to note that these models primarily focus on predicting complications that may arise during the hospitalization period, which can span from 24 hr up to 7 days. Therefore, when a patient is hospitalized, the utilization of Early Warning Scores (EWSs) is preferable as they predict the risk of complications within the next 24–48 hr. EWS enables standardized and daily evaluation of patients by detecting changes in clinical parameters that precede clinical deterioration (*Kostakis et al., 2021*; *Martín Rodríguez et al., 2021*).

Although EWS has been used in the evaluation of COVID-19 patients (*Aygun and Eraybar, 2022*; *Fang et al., 2021*; *Huespe et al., 2021*; *Liang et al., 2020*; *Sarkar et al., 2022*), there are clinical limitations to its use. For instance, the COVID-19 severity index (*Huespe et al., 2021*) employs vital signs, laboratory parameters, and chest X-ray findings to predict the risk of ICU admission within the next 24 hr. However, implementing these scoring systems requires the expertise of experienced healthcare professionals. The NEWS2 (*Royal College of Physicians, 2017*) also evaluates vital signs conducted by healthcare professionals, which may pose challenges during periods of overwhelming demand, and this model. Moreover, no models have been tested on geographically distinct populations or population groups with varying levels of immune protection.

In this study, we developed and validated, in a multicontinental cohort, COVID-19 Early Warning Score (COEWS): an EWS automatically calculated solely by laboratory parameters of widespread availability and affordability. Additionally, we evaluated the predictive performance of vaccinated and unvaccinated patients.

## Materials and methods

### Source of data

For the development and validation of this predictive model, we used data from the Hospital 12 de Octubre (Madrid, Spain) and the Argentinian COVID-19 Network (*Boietti et al., 2021*; *Casas-Rojo et al., 2020*). These registries comprise patients hospitalized between January 2020 and March 2022, with demographic, clinical, and analytical data.

The use of patient data in this study has received approval from the Clinical Research Ethics Committee of Hospital 12 de Octubre [reference 20/117]. Additionally, the Institutional Review Board of the Hospital Italiano de Buenos Aires approved the study for each participating site in the Argentinian COVID-19 Network (Approval Number: #5602). As the present study is a retrospective observational study, it does not involve any procedures or activities beyond the standard consultation and care provided to patients. It poses no additional risks to their health and incurs no extra costs for them or their healthcare coverage. Patient data will be assessed retrospectively through electronic medical records. Considering that this study involves minimal risk due to the handling of participant data, the corresponding ethics committees have granted an exemption from obtaining signed informed consent. This exemption is in accordance with Guideline 10: Modifications and Dispensations of Informed Consent outlined in the CIOMS 2019 guidelines.

### Participants

Patients hospitalized with a confirmed diagnosis of COVID-19 who received oxygen therapy were included. COVID infection was defined as a positive result of real-time reverse transcription-polymerase chain reaction for SARS-CoV-2 in nasopharyngeal swab specimens or sputum samples, also in the second and third waves, some patients were diagnosticated with the Panbio TM COVID-19 rapid test (Abbott) in the Spanish hospitals. We considered only the first hospitalization for COVID-19 for each patient during the study period. Patients were followed from hospital admission until death or hospital discharge.

### Outcome and variables

The aim of this study was the development of an EWS. Therefore, the outcome of the predictive model was death or mechanical ventilation (MV) in the next 48 hr. The potential predictors evaluated were: demographic data, admission vital signs (heart rate, temperature, systolic and diastolic blood pressure, and oxygen saturation), and blood test values (hemoglobin in g/dl, percentage lymphocytes, leukocytes in $10^3/mm^3$, platelets in $10^3/mm^3$, and glycemia mg/dl), vaccination status, and clinical variables that were present in all training and test cohorts. The percentage of missing values is listed in *Table 1*.

### Missing data

The used databases had some level of missingness due to the medical overload during the COVID-19 pandemic. In the training and internal testing database, we used simple imputation with the median values of the training dataset. For the external validation, we consider this missing data as 'Missing completely at random' (MCAR) (without systematic differences between the missing values and the observed values, *Sterne et al., 2009*), Hence, we performed multiple imputations by the Chained Equations procedure. To reduce the sampling error due to the imputations, we set the number of 20 imputed datasets (*White et al., 2011*).

### Sample size

To build a predictive model with approximately 10 estimated variables, we needed 10–20 outcome events per variable (*Katz, 2011*). We expected to include between 5 and 10 variables in the model, therefore we need at least 100–200 outcome events per database. Due to the fact that the number of outcomes (death and MV within 48 hr after hospital admission) in each database was higher than 200, we had enough sample size in the training and testing databases.

### COEWS development

The Spanish database was divided randomly into training (75%) and test (25%) datasets. COEWS was built using stable iterative variable selection (SIVS) (*Mahmoudian et al., 2021*) and linear regression

**Table 1.** Demographic characteristics, with missing data for each variable for the training database. We also include the coefficients of each variable included in the model.

| | Values in training cohort | Missing, n (%) | Coefficient |
|---|---|---|---|
| Sex (female) – n (%) | 3539 (47.1%) | 0 | 0.067 |
| Age in years – median (IQR) | 62.5 (47.8–78.7) | 0 | |
| Comorbidities | | | |
| Cardiac insufficiency – n (%) | 65 (0.8%) | 0 | |
| Peripheral vascular disease – n (%) | 365 (5.0%) | 0 | |
| Brain vascular disease – n (%) | 332 (4.5%) | 0 | |
| Dementia – n (%) | 415 (5.5%) | 0 | |
| COPD – n (%) | 592 (8.0%) | 0 | |
| Asthma – n (%) | 543 (7.0%) | 0 | |
| Diabetes – n (%) | 1525 (20.0%) | 0 | |
| Kidney disease – n (%) | 81 (1.0%) | 0 | |
| Liver disease – n (%) | 864 (11.5%) | 0 | |
| Solid tumor – n (%) | 1312 (17.5%) | 0 | |
| HIV – n (%) | 59 (0.7%) | 0 | |
| Active smoker – n (%) | 521 (7.0%) | 0 | |
| Obesity – n (%) | 1590 (21.0%) | 0 | |
| Clinical parameters at admission | | | |
| Temperature – Md (IQR) | 37.1 (36.5–37.9) | 53 (0.71) | |
| SBP– Md (IQR) | 127 (113–142) | 104 (1.39) | |
| Heart rate – Md (IQR) | 93 (80–106) | 76 (1.01) | |
| Respiration rate – Md (IQR) | 22 (18–28) | 5103 (67.98) | |
| O2_saturation (%) – Md (IQR) | 96 (93–98) | 102 (1.36) | 0.058 |
| Laboratory parameters | | | |
| Sodium (mmol/l) – Md (IQR) | 137 (134–139) | 264 (3.52) | |
| Potassium (mmol/l) – Md (IQR) | 4.12 (3.79–4.5) | 272 (3.62) | |
| Glucose (mg/dl) – Md (IQR) | 116 (101–142) | 306 (4.08) | 0.002 |
| Leukocytes ($\times 10^3/mm^3$) – Md (IQR) | 6.8 (5.1–9.3) | 245 (3.26) | |
| Neutrophils ($\times 10^3/mm^3$) – Md (IQR) | 4.9 (3.5–7.2) | 245 (3.26) | 0.068 |
| Percentage of lymphocytes – Md (IQR) | 15.6 (9.9–22.6) | 245 (3.26) | 0.008 |
| Hemoglobin (g/dl) – Md (IQR) | 13.8 (12.4–15.1) | 244 (3.26) | 0.044 |
| Platelets ($\times 10^3/mm^3$) – Md (IQR) | 210 (161–272) | 245 (3.26) | 0.0006 |
| Lactate (mmol/l) – Md (IQR) | 1.5 (1.2–2.2) | 7367 (98.14) | |
| Creatinine (mg/dl) – Md (IQR) | 0.9 (0.72–1.16) | 342 (4.56) | |
| LDH (U/l) – Md (IQR) | 330 (266–419) | 995 (13.2) | |
| GOT (U/l) – Md (IQR) | 34 (25–51) | 737 (9.82) | |
| GPT (U/l) – Md (IQR) | 27 (17–46) | 453 (6.03) | |
| Bilirubine (mg/dl) – Md (IQR) | 0.5 (0.3–0.7) | 648 (8.63) | |
| aPTT – Md (IQR) | 30 (28–33) | 549 (7.31) | |

*Table 1 continued on next page*

*Table 1 continued*

|  | Values in training cohort | Missing, *n* (%) | Coefficient |
|---|---|---|---|
| Prothrombin activity (%) – Md (IQR) | 83 (74–93) | 545 (7.26) | |
| Intercept | | | 4.235 |

Model calculation $C = A0 + A1* \times 1 + A2* \times 2 + A3* \times 3 + A4* \times 4 + A5* \times 5 + A6* \times 6 + A7* \times 7$. Example $4.23588\ldots + -0.05808\ldots*7.5 + \ldots -0.06734*1 = -0.280637$. aPTT: Activated Partial Thromboplastin Time, COPD: Chronic Obstructive Pulmonary Disease, GOT: glutamic-oxaloacetic transaminase, GPT: Glutamic-Pyruvic Transaminase, HIV : Human Immunodeficiency Virus, IQR: Intercuartile range, LDH: Lactate Dehydrogenase, SBP: Systolic Blood Pressure.

with the least absolute shrinkage and selection operator (Lasso) regularization (*Friedman et al., 2010*). In model building, only the training cohort was used and models were built using 10-fold cross-validation. In the feature selection stage of SIVS, 100 models were built and for each model selected variables were recorded. To reduce the number of features to as few as possible (therefore, increasing the easiness of use of COEWS), we used the weighting function in SIVS (called variable importance scoring) with a threshold of 0.15. This method has been shown to be very efficient, especially when the ratio of positive and negative outcomes is imbalanced (*Klén et al., 2019*). Lasso models were built in R Development Core Team, 2010 (version 3.6.0) package glmnet (*Friedman et al., 2010*) (version 4.1–1). All predictions were blinded to the final clinical outcome. The Argentinian database was used for external validation.

## Transformation of linear predictors to the score

Based on the final linear Lasso model and clinical insights, the final COEWS score was created. The clinical insights was used to determine the normal range, which gives zero points in the COEWS score. The other scoring ranges with negative and positive points were determined using the coefficients of the final Lasso model by linear interpolation from the COEWS. Especially, the ranges per feature were defined by the magnitude of the corresponding coefficient, and larger absolute values of coefficients yielded shorter ranges.

## Discrimination and calibration

The performance of the developed models was evaluated using the area under the receiver operating characteristic curve (AUROC), and the model's calibration was evaluated using the root mean square

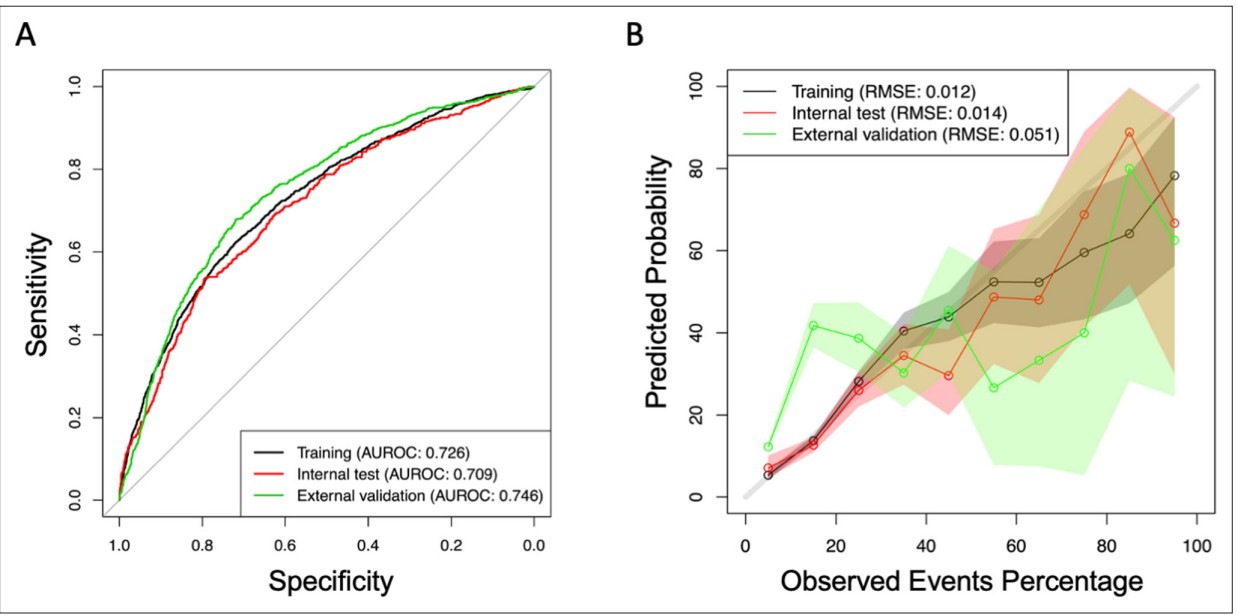

**Figure 1.** Area under the receiver operating characteristic curve (AUROC), (**A**) and calibration slopes for training and testing database (**B**).

error. We also evaluate the calibration by comparing the prediction of the model with the observed proportion of the combined outcome, stratifying every patient according to the probability predicted by the model in deciles. We performed a graph with the average probability of the outcome predicted by the model (average of individual estimated probabilities) and the observed probability (proportion of nonattendance) within each decile stratum (*Figure 1*; *Giunta et al., 2023*).

In order to account for the transformation of model coefficients into a score, we conducted additional evaluations to assess discrimination in each database using the score derived from the coefficients. Furthermore, we performed separate analyses for vaccinated and unvaccinated patients, as well as patients hospitalized before and after 2021, to investigate potential differences in predictive performance following changes in therapeutics. To determine the cutoff point for the analysis, we selected July 2021. This decision was based on the publication and implementation of the Recovery Trial indications, which introduced significant therapeutic changes during the pandemic (*Horby et al., 2021*). Lastly, we calculated the discrimination of the NEWS2 score to provide a comparative assessment with the COEWS score.

## Role of the funding source

The data collection for this article was supported by the Instituto de Salud Carlos III, the Ministry of Science and Innovation of Spain (COVID-19 COV20/00181) — co-financed by the European Development Regional Fund A way to achieve Europe and the Ministry of Health of Argentina (Becas Salud Investiga). The publication of this article was supported by the University of Vienna.

## Results

### Participants

Between January 2020 and February 2022, we included 15,903 hospital admissions of COVID-19 patients from the Spanish database (1009 patients) and the Argentinian database (5894 patients). The training database included 7507 patients and in the internal testing 2502 patients from the Hospital 12 de Octubre (Madrid, Spain). Of these, 471 (4.7%) were vaccinated with at least one COVID-19 vaccine dose (267 [2.7%] patients had a single dose, and 203 [2%] were fully vaccinated).

The external testing included 5894 patients from the Argentinian COVID-19 Network study, of these 669 (11.4%) were vaccinated with at least one COVID-19 vaccine dose (427 [7.2%] patients vaccinated with one COVID-19 vaccine dose and 242 [4.1%] with two).

**Table 2.** Scores of the COVID-19 Early Warning Score (COEWS) predictive model. Green color means 0 points, yellow 1 point, ornge 2 points, and red 3 or more points.

| Parameters | 6 | 5 | 4 | 3 | 2 | 1 | 0 | 1 | 2 | 3 | 4 | 5 | 6 |
|---|---|---|---|---|---|---|---|---|---|---|---|---|---|
| SpO$_2$ (%) | <85 | 85–86 | 87–88 | 89–90 | 91–92 | 93–95 | >95 | | | | | | |
| Neutrophils (×10³/mm³) | | | | | | <1.5 | 1.5–8.0 | 8.1–9.0 | 9.1–10.0 | 10.1–11.0 | 11.1–12.0 | 12.1–13.0 | >13 |
| Hemoglobin (g/dl) | | | | | <9 | 10–11 | 12–17 | 18–19 | >19 | | | | |
| Platelets (×10³/mm³) | | | | | | <150 | 150–400 | >400 | | | | | |
| Lymphocytes (%) | | | | | | <8 | 8–20 | >20 | | | | | |
| Glucose (mg/dl) | | | | | | <90 | 90–140 | 141–187 | 188–234 | 235–280 | >280 | | |
| Sex | | | | | | Male | Female | | | | | | |

| Global score |
|---|
| Low risk 0–3 |
| Moderate risk 4–5 |
| High risk 6–7 |
| Critical risk >7 |

The observed number of combined events (death and MV within 48 hr after admission) was 1477 (19.7%) in the training database, 452 (18.1%) in the test database, and 906 (15.4%) in the external validation database. In *Table 1*, we presented the demographic, laboratory, vital signs, and comorbidities of the training database.

## COEWS development

The variables of the COEWS predictive model were selected based on the coefficients but also based on the wide availability of these laboratory variables. Thus, the final model included features from the complete blood count, blood glucose, and oxygen saturation. We observed that the male sex, lower oxygen saturation, neutrophils, hemoglobin, platelets, lymphocytes, and glucose were positively correlated with death or MV in the next 48 hr. The coefficients of each variable included in the final model are presented in *Table 1*.

In order to make COEWS more actionable in real clinical situations, we transformed the predictive coefficients of the COEWS model into individual scores for each selected feature (*Table 2*). The global score, obtained by summing the individual scores, serves as an easy-to-calculate measure indicating the risk of a patient developing the combined outcome of MV or death within the next 48 hr. We established the levels of risk in the final score based on the NEWS2 Scale: lower risk (less than 10%), moderate risk (10–20%), high risk (20–30%), and critical risk (higher than 30%). Our aim was to categorize patients into different risk categories based on their likelihood of experiencing the combined outcome. To provide a comprehensive understanding of the outcome percentages for each risk level, we examined our cohort and found the following rates:

- Patients classified as low risk had an 8.5% rate of experiencing the outcome.
- Patients classified as moderate risk had an 18.4% rate of experiencing the outcome.
- Patients classified as high risk had a 25.6% rate of experiencing the outcome.
- Patients classified as critical risk had a 43.4% rate of experiencing the outcome.

These percentages illustrate the increasing likelihood of the outcome as the risk level escalates.

## Calibration and discrimination

The AUROC calculated for the training, testing, and external validation of the databases is presented in *Table 3*. The discrimination in the external validation cohort was 0.743 (0.703–0.784) for the COEWS score performed with coefficients and 0.700 (0.654–0.745) for the COEWS performed with the scores. Of note, the AUROC of vaccinated and non-vaccinated patients was similar. Additionally, we observed that the AUROC of the NEWS2 was 0.677 (0.601–0.752) in vaccinated patients and 0.648 (0.608–0.689) in unvaccinated patients (*Table 3*).

To account for changes in therapeutics, we performed an additional validation using patients hospitalized after July 2021. The model demonstrated similar discrimination ability, with an AUROC of 0.718 (0.569–0.867) for the COEWS score performed with coefficients in unvaccinated and 0.677 (0.612–0.742) for the COEWS score performed with coefficients in vaccinated (*Table 3*).

Furthermore, we observed a good and moderate calibration for the training and internal validation, and for the external validation cohort, respectively (*Figure 1*, *Supplementary file 1*, and *Supplementary file 2*).

**Table 3.** Area under the receiver operating characteristic curve (AUROC) of the COVID-19 Early Warning Score (COEWS) predictive model calculated with the coefficients and with the score in vaccinated and unvaccinated patinetes.

| AUROC (95% CI) | Vaccinated | | Non-vaccinated | |
| --- | --- | --- | --- | --- |
| | EWS LASSO | EWS score | EWS LASSO | EWS score |
| Training (Spanish data 75%) | 0.753 (0.656–0.851) | 0.748 (0.659–0.838) | 0.721 (0.706–0.736) | 0.723 (0.709–0.738) |
| Internal validation (Spanish data 25%) | 0.712 (0.565–0.859) | 0.684 (0.513–0.855) | 0.704 (0.677–0.732) | 0.711 (0.685–0.738) |
| External validation (Argentinian data) | 0.743 (0.703–0.784) | 0.700 (0.654–0.745) | 0.767 (0.749–0.785) | 0.741 (0.723–0.759) |
| NEWS2 in all databases | 0.677 (0.601–0.752) | | 0.648 (0.608–0.689) | |
| Patients hospitalized after July 2021 | 0.718 (0.569–0.867) | 0.682 (0.508–0.856) | 0.677 (0.612–0.742) | 0.705 (0.646–0.764) |

## Discussion

In this study, we developed and validated the COEWS predictive model. The COEWS score predicts death or MV within the next 48 hr based on routine and widely available laboratory measurements. This EWS performs a simple and fast triage of COVID-19 hospitalized patients without the need of a physical examination, which greatly reduces the workload of the healthcare team. COEWS was initially developed in a European cohort of COVID-19 patients but also showed a good predictive performance in a very different external cohort of patients from South America. We also observed that COEWS had better predictive performance than the widely used NEWS2 score. In order to increase the clinical usefulness of COEWS, we transformed the coefficients of the predictive model and created an easy-to-use score table that can be used without an electronic calculator.

Although several predictive models of severity in COVID-19 patients have been published (*Miller et al., 2022*), they only predict whether the patient will have the outcome during hospitalization, regardless of when the patient will have the outcome (*Miller et al., 2022*), which limits their clinical use during a pandemic situation. In this context, EWS models offer the clinical advantage to predict the outcome of interest in a short time horizon (e.g., the next 24–48 hr). In the context of the COVID-19 pandemic, several EWS has been developed and used (NEWS2, NEWS2 with age, Modified Early Warning Score (MEWS), and the COVID-19 severity index) (*Colombo et al., 2021*; *Huespe et al., 2021*; *Liao et al., 2020*; *Royal College of Physicians, 2017*). However, each of them have clear limitations such as not having a time horizon for prediction in the development, lack of testing in vaccinated versus non-vaccinated patients, and not using MV as a clinical outcome. The latest is necessary because COVID-19 patients also die under MV, and also because the measurement of vital signs that are included in an EWS model, will be altered due to MV and hemodynamic support. In fact, we observed that NEWS2 has poor discrimination when considering this combined outcome (death or MV). In this way, the COEWS score has better discrimination, even in an external cohort of patients from a different continent.

The COEWS is the first EWS in which the predictive performance in vaccinated and unvaccinated patients has been compared. Even though hospitalized vaccinated patients have more favorable outcomes from unvaccinated patients (*Busic et al., 2022*; *Huespe et al., 2022*), we observed that the predictive performance was similar between patients with and without vaccines. This observation strongly suggests that hospitalized COVID-19 patients show similar clinical and laboratory manifestations when they are going to suffer serious adverse events, independently of their vaccination status.

Additionally, we found that the COEWS demonstrated good discrimination in patients hospitalized after July 2021. These findings underscore the model's robustness over time, affirming its potential as a valuable clinical tool. Despite the evolution of therapeutic approaches during the study period, it is important to note that the laboratory and clinical signs of deterioration associated with severe COVID-19 have remained consistent. This consistency further supports the relevance and reliability of the COEWS in identifying patients at risk of severe disease.

Finally, it is interesting to note that the proportion of vaccinated patients hospitalized in Europe was less than half of that in the Argentinean cohort. This difference may be due to a higher proportion of BIBP-CorV (Beijing Institute of Biological Products; Sinopharm) vaccinations in Argentina, which has been reported to have lower efficacy in reducing hospitalizations compared to the BNT162b2 (Pfizer) vaccine (*Al-Momani et al., 2022*). Our team's recent study also suggests slightly lower efficacy for BIBP-CorV (Beijing Institute of Biological Products; Sinopharm) and Gam-COVID-Vac (Sputnik) vaccines in reducing mortality among hospitalized patients compared to BNT162b2 (Pfizer) (*Huespe et al., 2022*). However, investigating the precise reasons behind the disparity in vaccinated hospitalized patients between Spain and Argentina falls beyond the scope of our study. Further research is needed to explore this intriguing finding.

Our study has unique strengths. First, we have developed an easy and fast triage tool for COVID-19 hospitalized patients that can classify the risk of the patients only using widely available blood tests and without the need for clinical evaluations. Thus, the COEWS score can be added to the Electronic Health Record to automatically calculate the risk of patient deterioration and for recommending the appropriate level of care within a time window of 24–48 hr. Also, COEWS was created with a big and broad (from 2020 to 2022) patient database and externally validated in a even bigger cohort of patients from another continent, which increases the confidence in COEWS performance. Importantly, we evaluated the predictive performance of the score with the coefficients but also with the score

transformation, as the transformation of the coefficients in a score makes it easier to use but can reduce the predictive performance. Finally, our study is the first one evaluating the predictive performance of an EWS score in vaccinated and unvaccinated patients.

This study also has limitations. First, the amount of missingness in some parameters can alter the performance of COEWS. To reduce this effect, we used the MCAR method, which enabled us to perform complete case analysis without selection bias. Also, we used populations from Spain and Argentina, which prompts questions about the validity of our model for non-European nor South American populations. Finally, the datasets of Hospital 12 de Octubre carry a misclassification of some vaccinated patients as some hospitalized patients could have been previously vaccinated in other centers before their hospitalization. However, it is important to note that the misclassification issue primarily applies to patients who were vaccinated toward the end of the cohort follow-up. During the initial vaccination campaigns, all patients in the Hospital 12 de Octubre dataset were indeed vaccinated at their own hospital. This hospital maintains a close cohort due to its coverage area in Madrid. Additionally, we want to emphasize that the misclassification problem did not occur in the Argentinian database, and the results obtained from the vaccinated cohort patients in both countries were extremely similar.

The extensive external validation, its high performance, its easiness to use and the positive benchmark in comparison with the widely used NEW2, positions COEWS as a new tool of reference for assisting clinical decisions and improving patient care in the upcoming pandemic waves.

## Additional information

### Funding

| Funder | Grant reference number | Author |
| --- | --- | --- |
| University of Vienna | | David Gómez-Varela |

The funders had no role in study design, data collection, and interpretation, or the decision to submit the work for publication.

### Author contributions

Riku Klén, Data curation, Software, Formal analysis, Writing - review and editing; Ivan A Huespe, Conceptualization, Data curation, Formal analysis, Supervision, Investigation, Methodology, Writing - original draft, Project administration; Felipe Aníbal Gregalio, Conceptualization, Supervision, Investigation, Writing - original draft, Writing - review and editing; Antonio Lalueza Lalueza Blanco, Miguel Pedrera Jimenez, Matias A Mirofsky, Bruno Boietti, Ricardo Gómez-Huelgas, Supervision, Investigation, Writing - review and editing; Noelia Garcia Barrio, Supervision, Investigation, Visualization; Pascual Ruben Valdez, Conceptualization, Supervision, Investigation, Writing - review and editing; José Manuel Casas-Rojo, Juan Miguel Antón-Santos, Conceptualization, Resources, Investigation; Javier Alberto Pollan, Conceptualization, Supervision, Project administration, Writing - review and editing; David Gómez-Varela, Conceptualization, Supervision, Funding acquisition, Investigation, Methodology, Writing - original draft, Project administration, Writing - review and editing

### Author ORCIDs

Riku Klén http://orcid.org/0000-0002-0982-8360
Ivan A Huespe http://orcid.org/0000-0003-3445-6981
Felipe Aníbal Gregalio http://orcid.org/0000-0003-3189-2382
Juan Miguel Antón-Santos http://orcid.org/0000-0003-3443-1100
David Gómez-Varela http://orcid.org/0000-0003-2502-9419

### Ethics

The SEMI-COVID-19 Registry and the COVID registries of 12 de Octubre and the Costa del Sol hospitals have been approved by the Provincial Research Ethics Committee of Malaga (Spain; C.I.F. number: 0-9150013-B). Institutional review boards approved each participating site in the Argentinian COVID-19 Network study (approval numbers: 1575, 5562, and 5606).

Decision letter and Author response
Decision letter https://doi.org/10.7554/eLife.85618.sa1
Author response https://doi.org/10.7554/eLife.85618.sa2

## Additional files

### Supplementary files
• Supplementary file 1. Original values plotted in the area under the receiver operating characteristic curve (AUROC) shown in *Figure 1*.
• Supplementary file 2. Original values plotted in the calibration curve shown in *Figure 1*.
• MDAR checklist

### Data availability
The databases used in this article are not freely available because they are the property of the '12 de Octubre University Hospital' from Spain and the 'Sociedad Argentina de Medicina' from Argentina. If any researcher wants to use this data, please send a message to either Dr. Antonio Lalueza (lalueza@hotmail.com) or to Dr. Ivan Alfredo Huespe (ivan.huespe@hospitalitaliano.org.ar) including a project proposal. The data will be only available for non-commercial proposals. All processed data has been made available in Supplementary file 1 and Supplementary file 2.

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
