## [Editor Report]

This is an important contribution where the authors describe the development of a new Early Warning System for patients with COVID-19, namely COEWS. In their convincing approach, they integrate sex, SpO2, complete blood count and blood glucose. A score equal to or higher than 6 represents a high risk for the admitted COVID-19 patient in need of oxygen to be intubated or die within 48 hours. The score is tested in a Spanish cohort and validated through a smaller Spanish cohort and a larger Argentinian one. It shows good performance, better than that of NEWS2. Hence the paper is relevant for clinicians.

---

## [Decision Letter]

**Decision letter after peer review:**

Thank you for submitting your article "Development and validation of an Early Warning System for COVID-19 patients in a multicontinental cohort: The COEWS (COVID-19 Early Warning Score) Score" for consideration by *eLife*. Your article has been reviewed by 2 peer reviewers, and the evaluation has been overseen by a Reviewing Editor and Jos van der Meer as the Senior Editor.

Essential revisions:

– The most important comment is on the vaccination status of the participants. The vaccination percentages are low in both cohorts. Remarkably, the vaccination rates of the participant patients from Argentina are almost double compared to the rates of the participant patients from Spain. This fact (and perhaps a possible interpretation) should be mentioned in the Discussion section. Despite these percentages, even more important is the issue described in the following two sentences of the manuscript: "Finally, the datasets of Hospital 12 de Octubre carry a misclassification of some vaccinated patients as some hospitalized patients could have been previously vaccinated in other centers before their hospitalization. The lack of a centralized data service makes it impossible to track these patients". These sentences, subconsciously, give the reader the message "We don't know the vaccination status of the patients of the Spanish cohort" because the uncertainty they contain practically 'cancels' the non-vaccination status of these patients. Theoretically, according to these sentences, all non-vaccinated patients from Hospital 12 de Octubre might have been previously to a different vaccination center.

– In the introduction, some already existing Early Warning Scores are mentioned (including NEWS2 and CODOP). Is it possible to provide a little more information about these scores and models? It could be quite useful to the reader, even in the form of a table.

– In the paragraph "Transformation of linear predictors to the Score" the final sentence mentions "The other scoring ranges with negative and positive points were determined using the coefficients of the final Lasso model by linear interpolation from the COEWS". Can you be more specific, please?

– In the following paragraph "Discrimination and calibration" a certain graph is described: "We performed a graph with the average probability of the outcome predicted by the model (average of individual estimated probabilities) and the observed probability (proportion of nonattendance) within each decile stratum. Is this the graph of Figure 2b? If so, can this information be included in the manuscript text?

– Upon the COEWS score as presented in Table 2: The scoring of SpO2 parameter: Why is the classification ">95%" considered as "abnormal"? Is it because, like the NEWS2 score, patients with hypercapnic respiratory insufficiency have (as "standard") a lower everyday measurement of SpO2? The scoring of Neutrophils (%) parameter: Does it refer to the absolute count of neutrophils (per μL)? (according to the values given in Table 2). The scoring of Platelets parameter: Please be a little more specific about the scaling of the presented platelets values (absolute count, in thousands). Unless typed incorrectly, why is the value of 401000 so significant in order to solely receive one point? (While values of 400000 and 402000 receive zero and two points respectively). The levels of the final score ("The global score", as mentioned): How are these levels defined? Or, to make this question clearer, what is the percentage of an unlikely event (death or intubation) on each level?

– Additionally, about the levels of the final score: Is the second level (score of 4-5) also ranked as "Low risk" or can it be classified as "Intermediate"? The color "coding" used (yellow instead of green) differentiates it from the first level (score of 0-3).

– Why did the authors include only hospitalized patients in need of oxygen?

– Nothing is mentioned about ICF or the waiver from patients whose data were used.

– Patient data had a range of 2 years (2020-2022). In this time period much has been learned and much has changed-improved in the treatment of COVID-19 patients. Although a sensitivity analysis between unvaccinated and vaccinated patients is welcome, I would perform a sensitivity analysis or even from the beginning split the cohorts according to the waves of the pandemic.

---

## [Author Response]

Essential revisions:– The most important comment is on the vaccination status of the participants. The vaccination percentages are low in both cohorts. Remarkably, the vaccination rates of the participant patients from Argentina are almost double compared to the rates of the participant patients from Spain. This fact (and perhaps a possible interpretation) should be mentioned in the Discussion section.

We appreciate the reviewer's insightful observation regarding the disparity in the proportion of hospitalized vaccinated patients between the Spanish and Argentinian cohorts. Indeed, in the Spanish cohort, the proportion was 4.7%, while in the Argentinian cohort, it was 11.4%. This discrepancy is an interesting observation, and we believe it may be attributed to a higher proportion of BIBP-CorV (Beijing Institute of Biological Products; Sinopharm) vaccinations in Argentina. It has been reported that BIBP-CorV has lower efficacy in reducing hospitalizations compared to the BNT162b2 (Pfizer) vaccine. (Al-Momani et al., 2022).

Furthermore, our recent study conducted by our team suggests slightly lower efficacy for BIBP-CorV and Gam-COVID-Vac (Sputnik) vaccines in reducing mortality among hospitalized patients compared to BNT162b2 (Pfizer). (Huespe et al., 2023). We will include these findings in the Discussion section as recommended by the reviewer. However, we acknowledge that investigating the precise reasons behind the disparity in vaccinated hospitalized patients between Spain and Argentina falls beyond the scope of our study. Further research is needed to explore this intriguing finding and delve deeper into the underlying factors.

Thank you for bringing up this important point, and we will address it accordingly in the revised manuscript.

Despite these percentages, even more important is the issue described in the following two sentences of the manuscript: "Finally, the datasets of Hospital 12 de Octubre carry a misclassification of some vaccinated patients as some hospitalized patients could have been previously vaccinated in other centers before their hospitalization. The lack of a centralized data service makes it impossible to track these patients". These sentences, subconsciously, give the reader the message "We don't know the vaccination status of the patients of the Spanish cohort" because the uncertainty they contain practically 'cancels' the non-vaccination status of these patients. Theoretically, according to these sentences, all non-vaccinated patients from Hospital 12 de Octubre might have been previously to a different vaccination center.

We acknowledge that there is some potential for misclassification in this variable, and we agree that it is a limitation of our study. However, it is important to note that the misclassification issue primarily applies to patients who were vaccinated toward the end of the cohort follow-up. During the initial vaccination campaigns, all patients in the Hospital 12 de Octubre dataset were indeed vaccinated at their own hospital. This hospital maintains a close cohort due to its coverage area in Madrid.

Additionally, we want to emphasize that the misclassification problem did not occur in the Argentinian database, and the results obtained from the vaccinated cohort patients in both countries were extremely similar.

We appreciate the reviewer's attention to this matter, and we will provide a more comprehensive explanation of the potential misclassification limitations in the revised manuscript.

– In the introduction, some already existing Early Warning Scores are mentioned (including NEWS2 and CODOP). Is it possible to provide a little more information about these scores and models? It could be quite useful to the reader, even in the form of a table.

We agree with the reviewer suggestion and we improved the description of these scores in the introduction section.

– In the paragraph "Transformation of linear predictors to the Score" the final sentence mentions "The other scoring ranges with negative and positive points were determined using the coefficients of the final Lasso model by linear interpolation from the COEWS". Can you be more specific, please?

Thank you for the comment. We have now complemented “The other scoring ranges with negative and positive points were determined using the coefficients of the final Lasso model by linear interpolation from the COEWS.” with clarification “Especially, the ranges per feature were defined by the magnitude of the corresponding coefficient, and larger absolute values of coefficients yielded shorter ranges.”

– In the following paragraph "Discrimination and calibration" a certain graph is described: "We performed a graph with the average probability of the outcome predicted by the model (average of individual estimated probabilities) and the observed probability (proportion of nonattendance) within each decile stratum. Is this the graph of Figure 2b? If so, can this information be included in the manuscript text?

We included this information in the manuscript.

– Upon the COEWS score as presented in Table 2: The scoring of SpO2 parameter: Why is the classification ">95%" considered as "abnormal"? Is it because, like the NEWS2 score, patients with hypercapnic respiratory insufficiency have (as "standard") a lower everyday measurement of SpO2? The scoring of Neutrophils (%) parameter: Does it refer to the absolute count of neutrophils (per μL)? (according to the values given in Table 2). The scoring of Platelets parameter: Please be a little more specific about the scaling of the presented platelets values (absolute count, in thousands). Unless typed incorrectly, why is the value of 401000 so significant in order to solely receive one point? (While values of 400000 and 402000 receive zero and two points respectively).

Thank you for bringing up these important points regarding the scoring of parameters in the COEWS score as presented in Table 2. We appreciate your observation, and we apologize for the errors in the table. We have carefully reviewed and made the necessary corrections.

Regarding the SpO2 parameter, a value of "95%" should indeed be classified as normal, and we have rectified this classification accordingly.

For the Neutrophils parameter, the unit of measure is "x 10³/mm³" or "per cubic millimeter," and we have clarified this in the revised table.

Regarding the Platelets parameter, the cutoff point for scoring is 400 x 10³/mm³ or 400,000 platelets per cubic millimeter. Any value above this cutoff receives 1 point. We have also made this clarification in the revised table.

We apologize for any confusion caused by these misprints and appreciate your attention to detail.

The levels of the final score ("The global score", as mentioned): How are these levels defined? Or, to make this question clearer, what is the percentage of an unlikely event (death or intubation) on each level?

The global score, obtained by summing the individual scores, serves as an easy-to-calculate measure indicating the risk of a patient developing the combined outcome of mechanical ventilation or death within the next 48 hours. We established the levels of risk in the final score based on the NEWS 2 Scale: lower risk (less than 10%), moderate risk (10 to 20%), high risk (20 to 30%), and critical risk (higher than 30%). Our aim was to categorize patients into different risk categories based on their likelihood of experiencing the combined outcome.

To provide a comprehensive understanding of the outcome percentages for each risk level, we examined our cohort and found the following rates:

Patients classified as Low risk had an 8.5% rate of experiencing the outcome.Patients classified as moderate risk had an 18.4% rate of experiencing the outcome.Patients classified as High risk had a 25.6% rate of experiencing the outcome.Patients classified as critical risk had a 43.4% rate of experiencing the outcome.

These percentages illustrate the increasing likelihood of the outcome as the risk level escalates.

Thank you for your inquiry, and we improve the explanation of the risk classification in the article.

– Additionally, about the levels of the final score: Is the second level (score of 4-5) also ranked as "Low risk" or can it be classified as "Intermediate"? The color "coding" used (yellow instead of green) differentiates it from the first level (score of 0-3).

We apologize for the confusion caused by the incorrect classification in the manuscript. As the reviewer correctly points out, a score of 4-5 should be classified as a "Moderate risk" rather than "Low risk."

– Why did the authors include only hospitalized patients in need of oxygen?

During the pandemic, particularly in its early stages, there was a tendency to hospitalize mild or asymptomatic patients solely for the purpose of isolation. However, it is important to note that hospitalization for this group of clinically stable patients is not the current standard practice, as hospitalized patients now typically require oxygen therapy. The requirement for oxygen therapy helps standardize the population, ensuring that only patients with moderate or severe COVID-19 are included. This approach aims to exclude patients who might have been admitted to the hospital for social reasons rather than medical necessity.

– Nothing is mentioned about ICF or the waiver from patients whose data were used.

We appreciate the reviewer's comment and agree that it is important to address the issue of informed consent for the use of patient data in our study. Given the retrospective nature of our study, which relied on secondary databases from Spanish and Argentinean registries, it is important to note that the study is exempt from the requirement for informed consent.

This exemption from obtaining informed consent is in accordance with Guideline 10:

Modifications and Dispensations of Informed Consent outlined in the Council for International Organizations of Medical Sciences (CIOMS) 2019 guidelines. Furthermore, the use of patient data has been approved by the Clinical Research Ethics Committee of Hospital 12 de Octubre [reference 20/117] for the Spanish cohort. Additionally, institutional review boards approved the use of data at each participating site in the Argentinian COVID-19 Network study, with approval numbers 1575, 5562, and 5606. We will ensure to include a clear statement regarding the exemption from informed consent and the ethical approvals obtained in the revised manuscript to address this important aspect. Thank you for bringing it to our attention.

– Patient data had a range of 2 years (2020-2022). In this time period much has been learned and much has changed-improved in the treatment of COVID-19 patients. Although a sensitivity analysis between unvaccinated and vaccinated patients is welcome, I would perform a sensitivity analysis or even from the beginning split the cohorts according to the waves of the pandemic.

We appreciate the reviewer's suggestion and recognize the importance of considering the changes and improvements in the treatment of COVID-19 patients over time. To address this concern, we performed an additional validation of the COWES score using data from patients hospitalized after July 2021, which aligns with the implementation of significant therapeutic changes as indicated by the publication of the Recovery Trial findings. (RECOVERY Collaborative Group et al., 2021).

Through this new analysis, we observed a similar predictive value of the COWES score, indicating its robustness across different time periods. Despite the evolving therapeutic approaches, it is noteworthy that the laboratory and clinical signs of deterioration associated with severe COVID-19 have remained consistent. Consequently, the clinical manifestations and predictive factors of severe disease observed in patients were consistent from 2020 to 2022.

Al-Momani H, Aldajah K, Alda’ajah E, ALjafar Y, Abushawer Z. 2022. Effectiveness of Pfizer/BioNTech and Sinopharm COVID-19 vaccines in reducing hospital admissions in prince Hamza hospital, Jordan. *Front Public Health* 10:1008521.

Huespe IA, Ferraris A, Lalueza A, Valdez PR, Peroni ML, Cayetti LA, Mirofsky MA, Boietti B, Gómez-Huelgas R, Casas-Rojo JM, Antón-Santos JM, Núñez-Cortés JM, Lumbreras C,

Ramos-Rincón J-M, Barrio NG, Pedrera-Jiménez M, Martin-Escalante MD, Ruiz FR,

Onieva-García MÁ, Toso CR, Risk MR, Klén R, Pollán JA, Gómez-Varela D. 2023. COVID19 vaccines reduce mortality in hospitalized patients with oxygen requirements: Differences between vaccine subtypes. A multicontinental cohort study. *J Med Virol* 95:e28786.

RECOVERY Collaborative Group, Horby P, Lim WS, Emberson JR, Mafham M, Bell JL, Linsell L, Staplin N, Brightling C, Ustianowski A, Elmahi E, Prudon B, Green C, Felton T,

Chadwick D, Rege K, Fegan C, Chappell LC, Faust SN, Jaki T, Jeffery K, Montgomery A, Rowan K, Juszczak E, Baillie JK, Haynes R, Landray MJ. 2021. Dexamethasone in Hospitalized Patients with Covid-19. *N Engl J Med* 384:693–704.